# Deriving Hyperparameter Scaling Laws via Modern Optimization Theory

Egor Shulgin[†]    Dimitri von Rütte[‡]    Tianyue H. Zhang[¶,§]

Niccolò Ajroldi[§]    Bernhard Schölkopf[§,‡]    Antonio Orvieto[§]

## Abstract

Hyperparameter transfer has become an important component of modern large-scale training recipes. Existing methods, such as $\mu$P, primarily focus on transfer between model sizes, with transfer across batch sizes and training horizons often relying on empirical scaling rules informed by insights from timescale preservation, quadratic proxies, and continuous-time approximations. We study hyperparameter scaling laws for modern first-order optimizers through the lens of recent convergence bounds for methods based on the Linear Minimization Oracle (LMO), a framework that includes normalized SGD, signSGD (approximating Adam), and Muon. Treating bounds in recent literature as a proxy and minimizing them across different tuning regimes yields closed-form power-law schedules for learning rate, momentum, and batch size as functions of the iteration or token budget. Our analysis, holding model size fixed, recovers most insights and observations from the literature under a unified and principled perspective, with clear directions open for future research. Our results draw particular attention to the interaction between momentum and batch-size scaling, suggesting that optimal performance may be achieved with several scaling strategies.

## 1 Introduction

Given the high cost and practical limitations of deriving empirical scaling rules, hyperparameter transfer informed by theory has become a research area of keen interest, with the most prominent method, $\mu$P [43], enabling learning rate transfer across model sizes. However, these results *typically require a fixed batch size, momentum, and training horizon*, leading practitioners to revert to empirical scaling rules, often guided by quadratic analyses [29], stochastic differential equation (SDE) approximations [12, 27], timescale preservation arguments [3, 28], or norm-based views [15].
In this theoretical study, inspired by recent empirical work on optimal hyperparameter scaling as the token budget increases [30, 31, 36, 44], we reexamine scaling laws using performance bounds from optimization theory, leveraging recent advances that extend beyond convex settings and Euclidean geometry [25]. Compared to Bu et al. [9], Schaipp et al. [34], who demonstrate surprising agreement between SGD performance on convex problems and LLM training, our LMO framework aligns theory more closely with modern optimization practice, employing Adam [24] and Muon [22].

We study how learning rate, batch size, and momentum determine the best achievable performance across training horizons and compute budgets at a fixed model scale. Our analysis yields explicit scaling predictions for key hyperparameters across training horizons, which have previously been observed only empirically or were not motivated by a unified, theoretically principled setup.

On the practical side, our results provide insights along the promising direction of momentum scaling, explored in the contemporary literature [14, 28]. On the theoretical side, our work (especially point (iii) above) opens up several directions for research on scaling theory under modified initialization and gradient noise assumptions (App. D). More directly, revised rates and insights can be derived by incorporating weight decay, learning-rate scheduling, and warmup into our analysis.

---

[§]ELLIS Institute Tübingen, MPI for Intelligent Systems, Tübingen AI Center, Tübingen, Germany
[†]King Abdullah University of Science and Technology (KAUST), Thuwal, Saudi Arabia
[‡]ETH Zürich, Zürich, Switzerland
[¶] Mila, Quebec AI Institute & Université de Montréal, Quebec, Canada

## 2 PRELIMINARIES

Consider the optimization problem $\min_{x \in \mathbb{R}^d} f(x)$, where we assume access to mini-batch estimates $g$ of the gradient $\nabla f(\cdot)$ [8], with $f$ potentially being non-convex.

**Optimizers.** Let $b \in \mathbb{N}_{>0}$ be the batch size. We denote by $g_b$ the stochastic gradient of loss $f$. Fix the stepsize (learning rate) $\eta > 0$ and momentum parameter $\beta := 1 - \alpha \in [0, 1)$. Following [33], consider a norm $\| \cdot \|$, and the Linear Minimization Oracle (LMO)[1] method[2]:

$$m^{k+1} = (1 - \alpha)m^k + \alpha\, g_b^k, \qquad x^{k+1} = x^k + \eta \arg \min_{\|d\| \leq 1} \langle m^{k+1}, d \rangle. \qquad (1)$$

Choosing $\| \cdot \|$ recovers: (i) Euclidean $\| \cdot \| = \| \cdot \|_2$: normalized SGD with momentum; (ii) $\| \cdot \| = \| \cdot \|_\infty$: signSGD with momentum [5]; (iii) spectral norm: Muon (orthogonalized update) [4, 10, 22].

SignSGD can be easily linked to Adam [24], both theoretically [2, 4] and performance-wise [32, 46]. Many works (e.g. $\mu$P derivations; 43) directly derive results using this approximation. **Convergence bounds.** Consider a fixed momentum $\beta = 1 - \alpha$, batch size $b$ and step size $\eta$, and run the algorithm for $K$ iterations. We assume that (1) the gradient noise variance $\mathbb{E}\|g_b - \nabla f\|_2^2$ is upper bounded by a constant $\sigma^2/b$, (2) the loss $f$ has $L$-Lipschitz gradients, with respect to the general $\| \cdot \|$ norm, (3) $f$ is lower bounded by $f^{\inf}$ and $\Delta_0 = f(x^0) - f^{\inf}$.

Let $\| \cdot \|$ be any norm, with dual norm $\| \cdot \|_\star$, Kovalev [25, Theorem 2] proves that :

$$\min_{1 \leq k \leq K} \mathbb{E}\left[\|\nabla f(x^k)\|_\star\right] \leq \frac{\Delta_0}{\eta K} + \frac{2\rho\sigma}{\alpha\sqrt{b}K} + 2\rho\sigma\sqrt{\frac{\alpha}{b}} + \frac{7L\eta}{2} + \frac{2L\eta}{\alpha}, \qquad (2)$$

where $\rho \geq 1$ is a norm equivalence constant (defined from $\|v\|_\star \leq \rho\|v\|_2$ which always holds in finite dimensional spaces), dependent on the chosen norm. The right-hand-side contains: (i) a purely deterministic optimization term $\Delta_0/(\eta K)$; (ii) momentum "burn-in" / averaging term $2\rho\sigma/(\alpha\sqrt{b}K)$; (iii) a noise floor term $2\rho\sigma\sqrt{\frac{\alpha}{b}}$; (iv) smoothness/trust-region error terms proportional to $\eta$, including an $\eta/\alpha$ coupling. For the star-convex case [25], the gradient norm can be lower bounded with functional suboptimality $f(x^k) - f(x^\star)$ connecting equation 2 to the loss.

## 3 DERIVATION OF SCALING LAWS

We study how the bound from equation 2 scales with step size $\eta$, momentum $1 - \alpha$, batch size $b$, step count $K$, and token budget $T := bK$. To keep expressions compact, define

$$C_1 := \Delta_0, \qquad C_2 := 2\rho\sigma, \qquad C_3 := 4L.$$

Also note that $\frac{7L}{2}\eta + \frac{2L}{\alpha}\eta \lesssim C_3\,\eta\left(1 + \frac{1}{\alpha}\right)$, up to constant factors.

**Proxy objective.** Define the following proxy (the right-hand-side of equation 2 up to constants):

$$\text{risk}_K(\eta, \alpha, b) := \boxed{C_1 \frac{1}{\eta K}} + \boxed{\frac{C_2}{\sqrt{b}} \cdot \frac{1 + \alpha^{3/2}K}{\alpha K}} + \boxed{C_3\,\eta\left(1 + \frac{1}{\alpha}\right)}, \qquad (3)$$

$$\text{risk}_T(\eta, \alpha, b) := \text{risk}_{Kb}(\eta, \alpha, b) = \boxed{C_1 \frac{b}{\eta T}} + \boxed{\frac{C_2}{\sqrt{b}} \cdot \frac{b + \alpha^{3/2}T}{\alpha T}} + \boxed{C_3\,\eta\left(1 + \frac{1}{\alpha}\right)}. \qquad (4)$$

### 3.1 OPTIMIZATION OF THE PROXY OBJECTIVE

We first consider a *fixed momentum*, independent of $(b, \eta, K)$. In the large-horizon regime $\alpha^{3/2}K \gg 1$, the burn-in part of the teal term in equation 3 is dominated, and we can use a simplified proxy:

$$\text{risk}_K(\eta, b) \approx C_1 \frac{1}{\eta K} + \tilde{C}_2 \frac{1}{\sqrt{b}} + \tilde{C}_3\eta, \qquad \text{risk}_T(\eta, b) \approx C_1 \frac{b}{\eta T} + \tilde{C}_2 \frac{1}{\sqrt{b}} + \tilde{C}_3\eta, \qquad (5)$$

where $\tilde{C}_2 := C_2\sqrt{\alpha}$ and $\tilde{C}_3 := C_3\left(1 + \frac{1}{\alpha}\right)$. We have the following result, proved in Appendix C.1.

---

[1] With a slight abuse of notation for $\arg\min$ denoting any element from the set.

[2] Also known as Unconstrained Stochastic Conditional Gradient method [11, 16, 18, 20, 33]

**Theorem 1 (Fixed momentum, large-horizon proxy)** *Fix $\alpha \in (0, 1]$ and consider equation 5.*

1. *(Iteration scaling.) For fixed $(K, b)$, with $K$ large, the proxy is minimized by*

$$\eta_K^\star(b) \propto K^{-1/2}, \qquad \text{risk}_K^\star(b) \propto K^{-1/2} + b^{-1/2}. \tag{6}$$

    *Thus at fixed $K$ (ignoring token cost), the optimal learning rate is batch size independent, and increasing $b$ improves the bound.*

2. *(Token-budget scaling.) For fixed large $T$, at a fixed batch size $b$, the optimal learning rate scales as* $\eta_T^\star(b) \propto b^{1/2}T^{-1/2}$ . *Moreover, the joint minimizer $(\eta_T^\star, b_T^\star)$ satisfies*

$$b_T^\star \propto T^{1/2}, \qquad \eta_T^\star(b_T^\star) \propto (b_T^\star)^{1/2}T^{-1/2} \propto T^{-1/4}, \qquad \text{risk}_T^\star \propto T^{-1/4}. \tag{7}$$

    *In particular, under a fixed token budget we find a non-trivial token-optimal batch size.*

Note that equation 6 shows that for *fixed batch size $b$*, optimization can saturate. As shown in equation 7, one can scale $b$ with $T$ to fix this issue. However, there is another option: as discussed in Cutkosky & Mehta [13] and Shulgin et al. [37] – scaling momentum has a similar effect.

**Theorem 2 (Fixed batch size, large horizon proxy)** *At a fixed batch size $b$ and momentum $\beta = 1 - \alpha$, the optimal learning rate scales with $T$ as* $\eta_T^\star(b, \alpha) \propto b^{1/2}\alpha^{1/2}T^{-1/2}$ . *A subsequent minimization w.r.t. $\alpha$ (at a fixed $T$ and $b$) then leads to*

$$\alpha_T^\star(b) \propto b \cdot T^{-1/2}, \qquad \eta_T^\star(b) = \eta_T^\star(b, \alpha_T^\star(b)) \propto b \cdot T^{-3/4}, \qquad \text{risk}_T^\star \propto T^{-1/4}. \tag{8}$$

The proof can be found in Appendix C.2. Our last result considers tuning learning rate, momentum and batch size jointly at a given token budget. The proof, to be found in Appendix C.3, is substantially more involved, since teal term in equation 3 cannot be dropped.

**Theorem 3 (Jointly tuned $(\eta, \alpha, b)$ under a fixed token budget)** *For large $T$, minimizing equation 4 over $\eta > 0$, $\alpha \in (0, 1]$, and $b \geq 1$ yields the asymptotic scalings*

$$b_T^\star \propto T^{1/6}, \qquad \eta_T^\star \propto T^{-7/12}, \qquad \alpha_T^\star \propto T^{-1/3}, \qquad \text{risk}_T^\star \propto T^{-1/4}. \tag{9}$$

*Moreover, these schedules are consistent with equation 8 after plugging in $b = b_T^\star$.*

## 3.2 Optimal asymptotics

While Theorem 3 gives the asymptotic minimizer of the exact proxy $\text{risk}_T$, all three tuning regimes attain the same $\text{risk}_T^\star \propto T^{-1/4}$ rate. The practical question is therefore whether following Theorem 3 materially improves over simpler scaling rules.

The answer is negative. We begin with a corollary shown in Appendix C.4.

**Corollary 1 (Momentum tuning and Batch size constraints)** *As the token budget $T \to \infty$, we have the following properties following directly from the proofs of Theorems 1 and 2.*

1. *Assume optimal tuning of batch size and learning rate for a fixed arbitrary momentum $\beta = 1 - \alpha$. We have that $\text{risk}_T^\star(\alpha) \propto (1 + \alpha)^{1/4} T^{-1/4}$. Hence, re-tuning $\alpha$ can only lead to a $2^{1/4} \approx 1.19$ improvement in the rate constant.*

2. *If instead the batch size is capped by hardware, $b \leq b_{\max}$, then optimal tuning of the learning rate at an arbitrary momentum $\beta = 1 - \alpha$ leads to $\text{risk}_T \propto \alpha^{1/2}b_{\max}^{-1/2}$ – a non-vanishing noise floor. Allowing $\alpha$ to decrease with $T$ (Theorem 2) removes this floor and restores $\text{risk}_T^\star \propto T^{-1/4}$ even at fixed batch size.*

The corollary shows that if batch-size growth is non-problematic, then tuning momentum affects the rate only marginally. Instead, if there is a limit on the maximum batch size, tuning momentum with the token budget becomes necessary to achieve optimality.

Next, since both scaling $b$ like $T^{1/6}$ and $T^{1/2}$ lead to optimal asymptotics (up to a constant $< 1.2$), it is natural to ask *whether other batch-size scaling laws can still achieve the optimal rate*. The answer is positive, with the caveat that some choices may lead to extremely fast (and likely numerically unstable) scaling of momentum or learning rates. The next result is shown in Appendix C.5.

**Corollary 2 (Several batch size scalings are near-optimal)** *For large $T$, consider minimizing equation 4 under the choice $b(T) = T^{\phi}$. If $\phi \leq 1/2$, then the choice $\alpha_T^*(b) \propto b(T) \, T^{-1/2}, \eta(T) \propto b(T) \, T^{-3/4}$ proposed in Theorem 2 leads to a rate $T^{-1/4}$. If instead $\phi \in (1/2, 1)$, then the maximum achievable rate is $\mathrm{risk}_T \propto b(T)^{1/2} \, T^{-1/2}$, slower than $T^{-1/4}$.*

Finally, we ask: *Why is that minimization of $\mathrm{risk}_T$ in Theorem 3 predicts such a specific scaling? What is special about it?* The answer relies on the particular nature of equation 4, comprising terms evolving at different speeds even after optimal tuning: the suboptimality landscape with respect to the variable $b$, once learning rate and momentum are tuned, is flat, as shown in Appendix D.2.

**Numerical verification.** We verify our derivations numerically[3] in App. E and Figures 2. While the scope of this paper is to introduce a new theoretical tool and demonstrate alignment with the literature, we directly verify some of our findings empirically in Figure 1 in the Appendix.

## 4 DISCUSSION AND CONNECTIONS WITH THE LITERATURE

Among our regimes, the fixed-momentum case is the most practically relevant, since momentum is usually kept near a default value while batch size is chosen mainly for throughput rather than optimization [7, 19, 36]. In this setting, Theorem 1 recovers several transfer rules observed in practice: at token budget $T$, the optimal learning rate scales as $\eta^{\star} \propto b^{1/2}T^{-1/2}$, matching square-root scaling with batch size and the decay of the optimal learning rate with training horizon [12, 27, 31]. The same proxy predicts a non-trivial token-optimal batch size $b^{\star} \propto T^{1/2}$ and a minimum-batch/minimum-step tradeoff, consistent with the hyperbolic relations reported in large-scale training [29, 42]. This is a genuine distinction from SGD, whose classical proxy yields linear learning-rate scaling $\eta^{\star} \propto bT^{-1/2}$ but no interior token-optimal batch size after tuning (see App. A).

When batch growth is constrained, momentum tuning becomes the main mechanism for preserving efficiency: at fixed $b$, Theorem 2 gives $\alpha^{\star} \propto bT^{-1/2}$ and $\eta^{\star} \propto bT^{-3/4}$, consistent with recent momentum-scaling proposals for Adam-like methods [13, 14, 28]. Retaining the burn-in term in the fully tuned proxy further selects the asymptotic schedules $b^{\star} \propto T^{1/6}$, $\alpha^{\star} \propto T^{-1/3}$, and $\eta^{\star} \propto T^{-7/12}$, although several nearby power laws remain near-optimal and attain the same $T^{-1/4}$ rate. A remaining tension is the learning-rate-increase phenomenon reported when both batch size and token budget are increased jointly [26, 42]: our globally optimal schedules still predict decreasing learning rates with budget, but positive fitted exponents can arise along particular batch-growth paths $b(T)$, under transfer constraints, or when the proxy assumptions fail (e.g., non-$b^{-1/2}$ noise scaling or heavy tails). A fuller discussion is deferred to Appendix A.

## 5 CONCLUSION

We presented a proxy-based framework for fixed-model hyperparameter scaling derived from recent LMO convergence bounds. It predicts how learning rate, batch size, and momentum should vary with training horizon and token budget, and recovers square-root learning-rate scaling, a token-optimal batch size, and practical transfer rules across horizons. Its main limitations are the constant-learning-rate, fixed-model, no-weight-decay setup, together with a remaining tension around reported learning-rate increases when batch size and token budget grow jointly: our optimal schedules generally predict decay with budget, so this may reflect transfer constraints, assumption mismatch, or looseness of the proxy. Extending the theory to richer schedules, model-size effects, and tighter large-model proxies is a natural next step.

---

[3]We compute $\mathrm{risk}_T$ ($C_1 = C_2 = C_3 = 1$) for $\eta \in (10^{-15}, 10^4)$, $b \in (1, 10^{15})$, $\alpha \in (10^{-10}, 1)$, $T \in (10^2, 10^{22})$. These are sampled log-uniformly: 100 values are picked in the ranges above.

## ACKNOWLEDGMENTS

Tianyue H. Zhang, Niccolò Ajroldi, Bernhard Schölkopf, and Antonio Orvieto acknowledge the financial support of the Hector Foundation. Tianyue H. Zhang acknowledges the support of the Natural Sciences and Engineering Research Council of Canada (NSERC). We are grateful for the support and great feedback by OpenEuroLLM, in particular by Jörg Franke, Aaron Klein and David Salinas.

The work described herein has been supported in part by the EC under the grant No. 101195233: OpenEuroLLM ![Open Euro LLM logo] .

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

# APPENDIX

## CONTENTS

## A    EXPANDED DISCUSSION AND CONNECTIONS WITH THE LITERATURE

Among our analyses, fixed momentum is the most practically relevant, as momentum is usually set to a standard value and rarely tuned in modern training pipelines [7, 19, 36], though such practice is slowly changing [28, 31, 44]. On the other hand, batch size is often chosen for hardware efficiency and throughput rather than optimization performance. In these settings, our fixed-batch-size scaling laws can provide the appropriate theoretical description of achievable performance (see App. B).

**Scaling learning rate with batch size and token transfer at a fixed batch size.** Malladi et al. [27] and Compagnoni et al. [12] provide theoretical evidence for the well-known square root scaling law: for adaptive methods, at a fixed token budget, scaling $b \mapsto \kappa b$ requires $\eta \mapsto \kappa^{1/2}\eta$. Recently, Mlodozeniec et al. [31] verified this law at scale, and also noted that (at a fixed batch size) the compute-budget (token-horizon) transfer requires scaling the learning rate down as $\eta \mapsto \eta\kappa^{-1/2}$ when the number of training tokens is increased by a factor $\kappa$. The same law is discussed by [9] in the context of convex problems trained with SGD. We note that *both these trends are predicted by our Theorem* 1: at a given token budget $T$, the optimal learning rate scales as $b^{1/2}T^{-1/2}$. Further, note that our work differs from [12, 27] in a crucial way: these works derive these scalings by matching statistical properties of the iterates distributions (i.e., discard the blue in equation 4), building up on the intuition around sharp minima, generalization, and critical batch size [21, 23, 35, 39]. Our comparisons, though recovering their scalings as a special case, concern the expected gradient-norm "performance" (which can be connected to the loss) and capture a finite training horizon.

**Optimal batch size scaling with token budget.** Empirically, with fixed momentum, the optimal batch size scaling is found to be anywhere between $T^{0.3271}$ and $T^{0.8225}$ (Table 3, first column), which is reasonably[4] consistent with our result $T^{1/2}$. Several works instead discuss the notion of critical batch size: *the threshold at which further increasing batch size no longer gives near-linear speedup in optimization steps.* Zhang et al. [44] shows empirically that the critical batch size scales primarily with data size rather than model size. In addition, they show theoretically (for SGD on linear regression) that in the bias-dominated, early-training regime, the critical batch size is $b = 1$; as training progresses and variance becomes dominant, the critical/efficient batch size increases. A similar behavior can also be seen in our Figure 2 for LMOs, where we see that the optimal (not the critical!) batch size ramps up at around $10^9$ tokens[5], and is initially one (see third panel).

**SGD: optimal batch size and linear learning rate scaling.** The framework of Kovalev [25] covers normalized steepest descent (LMO) methods, not vanilla SGD. However, it is possible to extend our methodology to SGD and derive scaling results from well-known bounds [17]. We present this in App. C.6: a simple argument yields that at a fixed batch size the optimal learning rate scales as $\eta^\star(b, T) = bT^{-1/2}$, in perfect agreement with the linear scaling suggestion by Jastrzebski et al. [21], Smith et al. [39]. Next, our results suggest that, in contrast to LMO methods such as Muon or SignSGD with momentum, *SGD does not have a non-trivial token-optimal batch size* (given an optimal step size). This is in agreement with recent empirical results, suggesting SGD (in contrast to Adam) always profits from an increased iteration budget at low token count [28, 41].

**Relation between batch size and step count.** Our scaling rules also predict that, under fixed momentum, achieving a certain target performance requires both a minimum batch size (Fig. 9) and a minimum step count (Fig. 10). Indeed, von Rütte et al. [42] reports a hyperbolic relation between step count and batch size that closely resembles our theoretical results, suggesting this may be a fundamental property of LMO optimizers. We build on this and *derive this hyperbolic relationship* in App. E. Notably, a similar relation exists for standard SGD, as empirically reported by McCandlish et al. [29], where a minimum step count and token budget are required to achieve a certain target performance, but no minimum batch size requirement exists.

**Momentum scaling with batch size.** A practical implementation of the idea of momentum adaptation as the batch size is scaled appears only in very recent work. Marek et al. [28] propose to scale the $\beta_2$ momentum parameter in Adam as follows: $b \mapsto \kappa b$ requires $\beta_2 \mapsto \beta_2^\kappa$. At the same time, Fer-

---

[4]Note that differences in exact exponents may arise from the indirect relationship between gradient norm-based metrics and loss. See Appendix D for a simple sensitivity analysis showing how deviations from the i.i.d. bounded-variance mini-batch model and other constraints can shift the predicted exponents.

[5]The precise value for this phase transition crucially depends on momentum, see App. E, and on the values of $C_1, C_2, C_3$ in equation 4.

bach et al. [14], Orvieto & Gower [32] suggest that the choice $\beta_1 = \beta_2$ in Adam leads (after tuning) to near-optimal performance, drawing a direct link to SignSGD with momentum parameter $\beta$. Our scaling result in Theorem 2 shows that, at a fixed token budget $T$ and a fixed batch size $b$, the optimal momentum $\beta = 1 - \alpha \approx 1 - bT^{-1/2}$. Hence, if $b$ is scaled by a factor $\kappa$, we get $\beta \approx 1 - \kappa bT^{-1/2}$. Our result is deeply connected to the strategy by Marek et al. [28], indeed note that as $T \to \infty$ we have $\beta^\kappa \approx \left(1 - bT^{-1/2}\right)^\kappa = 1 - \kappa bT^{-1/2} + \mathcal{O}\left(T^{-1}\right)$. A decreasing optimal momentum as the batch size increases is also found to be effective by Zhang et al. [44].

**Learning rate scaling under jointly increasing batch and token budget.** To start, we recall (see first paragraph in this section) that our results align with observations that, *at a fixed batch size*, the optimal learning rate should decrease as the training horizon grows (Theorem 1; Mlodozeniec et al. [31]). Yet, how does the optimal learning rate change as the *batch size is chosen to scale* with the token budget? The results of von Rütte et al. [42] and Li et al. [26] (see Table 3) suggest a trend that clashes[6] with our theory: in this setup, the optimal learning rates are found to increase with the token budget at the optimal batch size. Meanwhile, Theorem 1 predicts that if $b \propto T^{1/2}$, then $\eta^\star = T^{-1/4}$. Appendix D.3 shows that a positive fitted exponent can arise without contradicting Theorem 1 when one measures $\eta^\star$ along a batch-scaling path $b(T)$ (hence $K(T) = T/b(T)$), e.g. when the budget is increased mainly via larger batches (fewer additional steps) or when hyperparameters are transferred across budgets. Further, note that the bound in equation 2 rules out $\eta^\star(T) \to \infty$ for the constant-step proxy because of the $O(\eta)$ term, independent of batch size and momentum.

Appendix D additionally discusses how deviations from the idealized assumptions behind the proxy (e.g., non-$b^{-1/2}$ noise scaling, heavy tails, or norm-mismatched variance for LMO methods) can change the predicted exponents. Overall, this points to a clear direction for future investigation: determine which part of the remaining mismatch is due to (i) protocol constraints vs. (ii) assumption mismatch vs. (iii) looseness of the bound in equation 2 for modern large model training regimes.

## B    BUDGET TRANSFER SUMMARY

In this section, we give a practical summary of the takeaways provided by Theorems 1, 2, and 3.

It is often feasible to tune hyperparameters only at a relatively small token budget $T_0$, but then one wishes to run a much longer training at $T_1 \gg T_0$. Since the proxy bound $\text{risk}_T$ in equation 4 depends explicitly on the horizon $T = bK$, a naive reuse of the short-run optimum $(\eta_0, \alpha_0, b_0)$ at $T_1$ is generally suboptimal. A simple alternative is to (i) tune at $T_0$ (e.g. by grid search) within a chosen hyperparameter family, and (ii) *extrapolate* the best configuration to $T_1$ using the power-law scalings suggested by the proxy analysis. We consider four natural transfer regimes depending on whether batch size $b$ and momentum $\alpha$ are tuned.

In terms of the more common momentum coefficient $\beta = 1 - \alpha$, decreasing $\alpha$ as $T$ grows corresponds to increasing momentum $\beta \uparrow 1$. Regime (B) corresponds to the standard fixed-batch large-horizon tuning where the dominant terms of the proxy bound are balanced (see, e.g., [13, 37] for related large-horizon momentum scalings). Regimes (C)–(D) additionally allow batch size to change with $T$, which is what pins down a token-optimal $b(T)$.

**Contrast to SGD.** For non-convex Euclidean SGD, the classical bound yields $\eta^\star(T, b) \propto b/\sqrt{T}$ (before stability caps), so a simple budget transfer is $\eta_1 = \eta_0 \frac{b_1}{b_0}\sqrt{\frac{T_0}{T_1}}$ (and if $b_1 = b_0$, then $\eta_1 = \eta_0\sqrt{\frac{T_0}{T_1}}$). In this bound, batch size mainly trades iterations for parallelism [29], whereas the LMO bound contains additional batch-dependent terms that can induce a non-trivial token-optimal $b(T)$.

---

[6]Note that the comparison with empirical learning-rate scaling is subtle and influenced by factors such as model size. DeepSeek [6] reports a decrease in optimal learning rate as the token budget increases. However, since DeepSeek also scales model size with training horizon, this effect may largely reflect the lower learning rates required by larger models. In contrast, Li et al. [26] explicitly models the role of model size, and von Rütte et al. [42] uses $\mu$P initialization to decouple size from learning rate. Both decoupled analyses find a positive correlation between the token horizon and the learning rate.

| Regime | Tuned at $T_0$ | Transfer rule to $T_1$ | Comments |
|---|---|---|---|
| (A) fixed $b$, fixed $\alpha$ | $\eta$ | $b_1 = b_0, \quad \alpha_1 = \alpha_0,$ $\eta_1 = \eta_0 \left(\frac{T_0}{T_1}\right)^{1/2}$ | Safest; only rescales stepsize. |
| (B) fixed $b$, tuned $\alpha$ | $(\eta, \alpha)$ | $b_1 = b_0, \quad \alpha_1 = \alpha_0 \left(\frac{T_0}{T_1}\right)^{1/2},$ $\eta_1 = \eta_0 \left(\frac{T_0}{T_1}\right)^{3/4}$ | Fixed-batch large-horizon scaling ($\eta \propto K^{-3/4}, \alpha \propto K^{-1/2}$). |
| (C) tuned $b$, fixed $\alpha$ | $(\eta, b)$ | $\alpha_1 = \alpha_0, \quad b_1 = b_0 \left(\frac{T_1}{T_0}\right)^{1/2},$ $\eta_1 = \eta_0 \left(\frac{T_0}{T_1}\right)^{1/4}$ | Token-optimal $b$ under fixed $\alpha$ proxy (aggressive batch growth). |
| (D) tuned $b$, tuned $\alpha$ | $(\eta, \alpha, b)$ | $b_1 = b_0 \left(\frac{T_1}{T_0}\right)^{1/6},$ $\alpha_1 = \alpha_0 \left(\frac{T_0}{T_1}\right)^{1/3},$ $\eta_1 = \eta_0 \left(\frac{T_0}{T_1}\right)^{7/12}$ | Joint token-optimal proxy (burn-in term retained); milder batch growth. |

Table 1: Budget transfer rules for LMO methods under token budget $T = bK$. Here $(\eta_0, \alpha_0, b_0)$ is the best configuration found at $T_0$ within the chosen regime (e.g. via grid search), and $(\eta_1, \alpha_1, b_1)$ is the extrapolated configuration for $T_1$. All scalings are asymptotic and should be combined with feasibility constraints (e.g. $b_1 \geq 1$ integer, hardware caps, $\alpha_1 \in (0, 1]$, and stepsize stability limits).

| Setting | Calibrated invariants at $(T_0, b_0)$ | Extrapolation to $(T_1, b_1)$ |
|---|---|---|
| LMO, fixed $\alpha$ (Regime A) | $c_\eta := \eta_0 \sqrt{\frac{T_0}{b_0}}$ | $\eta_1 = c_\eta \sqrt{\frac{b_1}{T_1}} = \eta_0 \sqrt{\frac{b_1}{b_0}} \sqrt{\frac{T_0}{T_1}}$ |
| LMO, tuned $\alpha$ (Regime B) | $c_\alpha := \alpha_0 \frac{\sqrt{T_0}}{b_0}, \quad c_\eta := \eta_0 \frac{T_0^{3/4}}{b_0}$ | $\alpha_1 = c_\alpha \frac{b_1}{\sqrt{T_1}} = \alpha_0 \frac{b_1}{b_0} \sqrt{\frac{T_0}{T_1}}$ $\eta_1 = c_\eta \frac{b_1}{T_1^{3/4}} = \eta_0 \frac{b_1}{b_0} \left(\frac{T_0}{T_1}\right)^{3/4}$ |
| SGD | $c_\eta := \eta_0 \frac{\sqrt{T_0}}{b_0}$ | $\eta_1 = c_\eta \frac{b_1}{\sqrt{T_1}} = \eta_0 \frac{b_1}{b_0} \sqrt{\frac{T_0}{T_1}}$ |

Table 2: Budget transfer when batch size changes between short-run tuning at $(T_0, b_0)$ and long-run training at $(T_1, b_1)$. Here $(\eta_0, \alpha_0)$ are obtained by tuning at $(T_0, b_0)$ in the specified setting, and then extrapolated to $(T_1, b_1)$. All formulas are asymptotic and should be combined with feasibility constraints (e.g. $\alpha_1 \in (0, 1]$, integer batch sizes, hardware limits, and stepsize stability caps).

**Changing batch size between tuning and the long run.** Table 1 assumes the batch size is held fixed between tuning at $T_0$ and the long run at $T_1$. If instead the available hardware increases and one runs the long run with a larger batch, Table 2 summarizes the corresponding transfer rules (obtained by re-expressing the $K$-optimal schedules under $K(T) = T/b(T)$).

| | Batch Size | Learning Rate | Optimizer |
|---|---|---|---|
| DeepSeek [6] | $b^\star \propto T^{0.3271}$ | $\eta^\star \propto T^{-0.1250}$ | AdamW [24] |
| StepLaw [26] | $b^\star \propto T^{0.571}$ | $\eta^\star \propto N^{-0.713} T^{0.307}$ | AdamW [24] |
| von Rütte et al. [42] | $b^\star \propto T^{0.8225}$ | $\eta^\star \propto T^{0.2806}$ | LaProp [47] |

Table 3: Empirical batch size and learning rate scaling across three prior studies. These investigations use optimizers with fixed momentum, sweeping jointly over batch size and learning rates. $N$ denotes the number of non-embedding parameters.

## C  PROOFS

We provide here proofs for the results in Section 3.

### C.1  FIXED MOMENTUM, LARGE BUDGET

For a fixed $K$,

$$\mathrm{risk}_K(\eta, b) \sim C_1 \frac{1}{\eta K} + \tilde{C}_2 \frac{1}{\sqrt{b}} + \tilde{C}_3 \eta.$$

Minimizing w.r.t. $\eta$ gives

$$\eta_K^\star = \sqrt{\frac{C_1}{\tilde{C}_3 K}}.$$

Thus

$$\mathrm{risk}_K^\star \sim 2 \sqrt{\frac{C_1 \tilde{C}_3}{K}} + \tilde{C}_2 \frac{1}{\sqrt{b}},$$

and the performance therefore improves with $b$. $b_K^\star \to \infty$.

For a fixed $T$,

$$\mathrm{risk}_T(\eta, b) \sim C_1 \frac{b}{\eta T} + \tilde{C}_2 \frac{1}{\sqrt{b}} + \tilde{C}_3 \eta.$$

Minimizing w.r.t. $\eta$:

$$\eta_T^\star(b) = \sqrt{\frac{C_1 b}{\tilde{C}_3 T}}.$$

Plugging in and minimizing w.r.t. $b$ yields

$$b_T^\star = \frac{\tilde{C}_2}{2\sqrt{C_1 \tilde{C}_3}} \sqrt{T}, \qquad \eta_T^\star = \frac{C_1^{1/4} \tilde{C}_2^{1/2}}{\sqrt{2} \tilde{C}_3^{3/4}} \frac{1}{T^{1/4}}.$$

Moreover,

$$\mathrm{risk}_T^\star \sim 2\sqrt{2}(C_1 \tilde{C}_3)^{1/4} \tilde{C}_2^{1/2} \frac{1}{T^{1/4}}.$$

### C.2  FIXED BATCH SIZE, LARGE BUDGET

Write $\mathrm{risk}_K$ in expanded form:

$$\mathrm{risk}_K(\eta, \alpha, b) = C_1 \frac{1}{\eta K} + C_2 \frac{1}{\alpha \sqrt{b} K} + C_2 \sqrt{\frac{\alpha}{b}} + C_3 \eta \frac{1 + \alpha}{\alpha}.$$

In the large-horizon regime and at the optimizer (where $\alpha$ is small), we consider the leading proxy

$$\mathrm{risk}_K^{\mathrm{lead}}(\eta, \alpha, b) = C_1 \frac{1}{\eta K} + C_2 \sqrt{\frac{\alpha}{b}} + C_3 \frac{\eta}{\alpha}.$$

We check that this proxy is asymptotically correct at the end of the proof.

For fixed $(\alpha, b)$, minimization w.r.t. $\eta$ leads to

$$C_1 \frac{1}{\eta K} + C_3 \frac{\eta}{\alpha} \quad \Rightarrow \quad \eta^\star(\alpha, b, K) = \sqrt{\frac{C_1 \alpha}{C_3 K}}.$$

Substituting this gives

$$\min_{\eta > 0} \mathrm{risk}_K^{\mathrm{lead}} = 2 \sqrt{\frac{C_1 C_3}{K \alpha}} + C_2 \sqrt{\frac{\alpha}{b}}.$$

Next, we minimize w.r.t. $\alpha$. Let $p = 2\sqrt{\frac{C_1 C_3}{K}}$ and $q = \frac{C_2}{\sqrt{b}}$. Then we minimize $p\,\alpha^{-1/2} + q\,\alpha^{1/2}$, whose minimizer is $\alpha = p/q$, hence

$$\alpha_K^\star(b) = \frac{2\sqrt{C_1 C_3}}{C_2}\,\frac{\sqrt{b}}{\sqrt{K}}.$$

Plugging back yields

$$\min_{\alpha, \eta} \mathrm{risk}_K^{\mathrm{lead}} = 2\sqrt{pq} = 2\sqrt{2}\,C_2^{1/2}(C_1 C_3)^{1/4}\,\frac{1}{b^{1/4} K^{1/4}}.$$

Finally, using $\eta^\star(\alpha, b, K) = \sqrt{\frac{C_1 \alpha}{C_3 K}}$ with $\alpha = \alpha_K^\star(b)$ gives

$$\eta_K^\star(b) = \sqrt{2}\,\frac{C_1^{3/4}}{C_2^{1/2} C_3^{1/4}}\,\frac{b^{1/4}}{K^{3/4}}.$$

For $b = 1$ this recovers $\alpha \propto K^{-1/2}$ and $\eta \propto K^{-3/4}$.

**Consistency of dropping lower-order terms.** At $(\eta_K^\star, \alpha_K^\star)$, the dropped burn-in term scales as

$$C_2 \frac{1}{\alpha\sqrt{b}\,K} = \mathcal{O}\!\left(\frac{1}{b\,K^{1/2}}\right),$$

while the dropped additive smoothness term $C_3 \eta$ scales as $\mathcal{O}(b^{1/4} K^{-3/4})$, both lower order than the leading $\mathcal{O}(b^{-1/4} K^{-1/4})$ term for large $K$.

**Remark 4 (Continuum of feasible batch-growth paths under Theorem 2)** *Although Theorem 2 is stated for fixed $b$, the schedules are explicit in $b$: $\alpha_T^\star(b) \propto bT^{-1/2}$ and $\eta_T^\star(b) \propto bT^{-3/4}$ with $\mathrm{risk}_T^\star \propto T^{-1/4}$. Therefore, along any path $b(T)$ such that $\alpha_T^\star(b(T)) \leq 1$ (equivalently $b(T) \lesssim \sqrt{T}$), one retains the same token exponent $T^{-1/4}$, while the induced scalings of $\alpha$ and $\eta$ depend on the chosen $b(T)$. This "continuum" is broken (and a specific $b_T^\star$ is selected) once the burn-in term is retained, as in Theorem 3.*

## C.3 FULL TUNING, LARGE BUDGET

The following discussion is self-contained and general enough to also describe the results we presented in simpler settings, as we show after the proof.

**Convergence bound.** We start from the non-convex bound (up to universal constants)

$$\min_{1 \leq k \leq K} \mathbb{E}\big[\|\nabla f(x^k)\|_\star\big] \;\leq\; \underbrace{\frac{\Delta_0}{\eta K} \;+\; \frac{2\rho\sigma}{\alpha\sqrt{b}\,K} \;+\; 2\rho\sigma\sqrt{\frac{\alpha}{b}} \;+\; \frac{7L\eta}{2} \;+\; \frac{2L\eta}{\alpha}}_{=:\,\mathcal{U}(\alpha, \eta; b, K)}. \qquad (10)$$

Here $\eta > 0$ is the step size, $\alpha \in (0,1]$ is the momentum "update" parameter ($\beta = 1 - \alpha$), $b$ is the batch size, and $K$ is the number of iterations.

**Token budget.** Fix a total budget $T = bK$. Substitute $K = T/b$ into $\mathcal{U}(\alpha, \eta; b, K)$ and define

$$\mathcal{U}_T(\alpha, \eta; b) := \mathcal{U}(\alpha, \eta; b, T/b) = \frac{b\Delta_0}{\eta T} + \frac{2\rho\sigma\sqrt{b}}{\alpha T} + 2\rho\sigma\sqrt{\frac{\alpha}{b}} + \frac{7L\eta}{2} + \frac{2L\eta}{\alpha}. \qquad (11)$$

We now minimize equation 11 over $(\eta, \alpha, b)$ (treating $b$ as a continuous variable; in practice $b$ is an integer and must respect hardware caps).

**Step 1: minimize over $\eta$ (for fixed $\alpha$, b).** For fixed $(\alpha, b)$, the $\eta$-dependent part of equation 11 is

$$\frac{b\Delta_0}{T} \cdot \frac{1}{\eta} + L\left(\frac{7}{2} + \frac{2}{\alpha}\right)\eta.$$

This function, with respect to $\eta$, is easy to minimize. The minimizer is

$$\eta_T^{\star}(\alpha, b) = \sqrt{\frac{b\Delta_0}{T\,L\left(\frac{7}{2} + \frac{2}{\alpha}\right)}} \propto \sqrt{b/T}, \qquad \min_{\eta>0}\left\{\frac{A}{\eta} + B\eta\right\} = 2\sqrt{AB}. \tag{12}$$

Plugging equation 12 into equation 11 yields the $\eta$-optimized bound

$$\min_{\eta>0}\mathcal{U}_T(\alpha, \eta; b) = 2\sqrt{\frac{b\Delta_0 L}{T}\left(\frac{7}{2} + \frac{2}{\alpha}\right)} + \frac{2\rho\sigma\sqrt{b}}{\alpha T} + 2\rho\sigma\sqrt{\frac{\alpha}{b}} =: \Phi_T(\alpha, b). \tag{13}$$

**Step 2: minimize $\Phi_T(\alpha, b)$ over $b$ (for fixed $\alpha$).** Let $s := \sqrt{b} > 0$. Then equation 13 can be written as

$$\Phi_T(\alpha, b) = A_T(\alpha)\,s + \frac{B(\alpha)}{s}, \tag{14}$$

where

$$A_T(\alpha) = 2\sqrt{\frac{\Delta_0 L}{T}}\sqrt{\frac{7}{2} + \frac{2}{\alpha}} + \frac{2\rho\sigma}{\alpha T}, \qquad B(\alpha) = 2\rho\sigma\sqrt{\alpha}. \tag{15}$$

Since $As + B/s$ is minimized at $s^{\star} = \sqrt{B/A}$, we obtain

$$\sqrt{b_T^{\star}(\alpha)} = \sqrt{\frac{B(\alpha)}{A_T(\alpha)}}, \qquad b_T^{\star}(\alpha) = \frac{B(\alpha)}{A_T(\alpha)} = \frac{2\rho\sigma\sqrt{\alpha}}{2\sqrt{\frac{\Delta_0 L}{T}}\sqrt{\frac{7}{2} + \frac{2}{\alpha}} + \frac{2\rho\sigma}{\alpha T}}. \tag{16}$$

The corresponding minimized value (for fixed $\alpha$) is

$$\min_{b>0}\Phi_T(\alpha, b) = 2\sqrt{A_T(\alpha)\,B(\alpha)}. \tag{17}$$

Using $\sqrt{\alpha}\sqrt{\frac{7}{2} + \frac{2}{\alpha}} = \sqrt{\frac{7}{2}\alpha + 2}$, we can simplify:

$$\min_{b>0}\Phi_T(\alpha, b) = 4\sqrt{\rho\sigma\sqrt{\frac{\Delta_0 L}{T}}\sqrt{\frac{7}{2}\alpha + 2} + \frac{(\rho\sigma)^2}{T\sqrt{\alpha}}} =: \Psi_T(\alpha). \tag{18}$$

**Step 3: minimize $\Psi_T(\alpha)$ over $\alpha$ (exact cubic condition).** Because the outer square-root in equation 18 is monotone, minimizing $\Psi_T(\alpha)$ is equivalent to minimizing the inner expression

$$g_T(\alpha) := \rho\sigma\sqrt{\frac{\Delta_0 L}{T}}\sqrt{\frac{7}{2}\alpha + 2} + \frac{(\rho\sigma)^2}{T\sqrt{\alpha}}. \tag{19}$$

Differentiate:

$$g_T'(\alpha) = \rho\sigma\sqrt{\frac{\Delta_0 L}{T}} \cdot \frac{\frac{7}{2}}{2\sqrt{\frac{7}{2}\alpha + 2}} - \frac{(\rho\sigma)^2}{T} \cdot \frac{1}{2}\alpha^{-3/2}.$$

Setting $g_T'(\alpha) = 0$ and rearranging gives

$$\rho\sigma\sqrt{\frac{\Delta_0 L}{T}} \cdot \frac{\frac{7}{2}}{\sqrt{\frac{7}{2}\alpha + 2}} = \frac{(\rho\sigma)^2}{T}\alpha^{-3/2}.$$

Squaring both sides (valid for $\alpha > 0$) yields the *exact cubic*:

$$\left(\frac{7}{2}\right)^2\Delta_0 L\,T\,\alpha^3 - \left(\frac{7}{2}\right)(\rho\sigma)^2\alpha - 2(\rho\sigma)^2 = 0. \tag{20}$$

This equation has a unique positive real root; denote it by $\alpha_T^{\star}$. Then $\alpha_T^{\star}$ is the unique positive root of equation 20, $b_T^{\star} = b_T^{\star}(\alpha_T^{\star})$ from equation 16, $\eta_T^{\star} = \eta_T^{\star}(\alpha_T^{\star}, b_T^{\star})$ from equation 12, $K_T^{\star} = \frac{T}{b_T^{\star}}$.

$$\alpha_T^{\star} = \text{the unique positive root of equation 20},$$
$$b_T^{\star} = b_T^{\star}(\alpha_T^{\star}) \text{ from equation 16},$$
$$\eta_T^{\star} = \eta_T^{\star}(\alpha_T^{\star}, b_T^{\star}) \text{ from equation 12}, \quad K_T^{\star} = T/b_T^{\star}.$$

**Asymptotic scalings for large** $T$**.** We now extract an explicit approximation for $\alpha_T^\star$ from equation 20. Write the cubic in the condensed form

$$A\,T\,\alpha^3 \;-\; B\,\alpha \;-\; C \;=\; 0, \qquad A := \left(\frac{7}{2}\right)^2 \Delta_0 L, \; B := \left(\frac{7}{2}\right)(\rho\sigma)^2, \; C := 2(\rho\sigma)^2. \tag{21}$$

STEP (A): IDENTIFY THE CORRECT EXPONENT. Assume $\alpha$ decays polynomially, $\alpha \propto T^{-p}$. Then the three terms scale as

$$A T \alpha^3 \propto T^{1-3p}, \qquad B\alpha \propto T^{-p}, \qquad C \propto T^0.$$

To balance the *constant* term $C$ with the leading term $AT\alpha^3$ we require $1 - 3p = 0$, hence $p = \frac{1}{3}$. This predicts $\alpha_T^\star = \Theta(T^{-1/3})$.

STEP (B): COMPUTE THE LEADING CONSTANT. Set the rescaled variable $u := \alpha\,T^{1/3}$, i.e. $\alpha = u\,T^{-1/3}$. Plugging into equation 21 gives

$$A T \left(u^3 T^{-1}\right) \;-\; B\left(u T^{-1/3}\right) \;-\; C \;=\; 0 \quad \Longleftrightarrow \quad A u^3 \;-\; C \;=\; B u\,T^{-1/3}.$$

As $T \to \infty$, the right-hand side vanishes, so $u$ converges to the unique positive root of $Au^3 - C = 0$, i.e. $u_0 = (C/A)^{1/3}$. Therefore,

$$\alpha_T^\star \;\approx\; u_0\,T^{-1/3} = \left(\frac{C}{A}\right)^{1/3} T^{-1/3} = \left(\frac{2(\rho\sigma)^2}{\left(\frac{7}{2}\right)^2 \Delta_0 L}\right)^{1/3} T^{-1/3} = \frac{2}{7^{2/3}}\frac{(\rho\sigma)^{2/3}}{(\Delta_0 L)^{1/3}}\,T^{-1/3}. \tag{22}$$

FIRST CORRECTION TERM. The same rescaling gives $Au^3 - C = Bu\,T^{-1/3}$, so one may expand $u = u_0 + u_1 T^{-1/3} + \mathcal{O}(T^{-2/3})$. Keeping the order-$T^{-1/3}$ terms yields $3Au_0^2 u_1 = Bu_0$, hence

$$u_1 = \frac{B}{3Au_0} = \frac{B}{3A^{2/3}C^{1/3}}, \quad \text{and thus} \quad \alpha_T^\star = u_0 T^{-1/3} + u_1 T^{-2/3} + \mathcal{O}(T^{-1}).$$

STEP (C): INDUCED BATCH SIZE AND STEP-SIZE SCALINGS. Using equation 16 and the fact that $\alpha_T^\star \to 0$, we have $\frac{7}{2} + \frac{2}{\alpha} \sim \frac{2}{\alpha}$, and also the term $\frac{2\rho\sigma}{\alpha T}$ in $A_T(\alpha)$ becomes lower order at $\alpha = \alpha_T^\star$. A short calculation then yields the scalings

$$b_T^\star = \Theta(T^{1/6}), \qquad K_T^\star = \Theta(T^{5/6}), \qquad \eta_T^\star = \Theta(T^{-7/12}), \tag{23}$$

and substituting the optimized parameters back into equation 18 gives the rate

$$\min_{1 \le k \le K_T^\star} \mathbb{E}\big[\|\nabla f(x^k)\|_\star\big] = \mathcal{O}\Big((\Delta_0 L)^{1/4}\sqrt{\rho\sigma}\,T^{-1/4}\Big).$$

**Recovering earlier regimes as special cases.** All previously discussed tuning regimes are obtained by *restricting* the optimization above:

- *Fixed $b$ (no batch tuning):* keep $b$ fixed in equation 11 and optimize only over $(\eta, \alpha)$ (equivalently, apply Steps 1 and 3 but without Step 2). This recovers the familiar large-horizon schedules $\alpha \propto K^{-1/2}$ and $\eta \propto K^{-3/4}$ (up to $b$-dependent constants) once one rewrites $K = T/b$.

- *Fixed $\alpha$ (no momentum tuning):* keep $\alpha$ fixed and optimize over $(\eta, b)$. Then Step 3 is skipped and the minimizer in Step 2 yields the "fixed-$\alpha$" token-optimal batch scaling (typically $b_T^\star = \Theta(\sqrt{T})$ in the simplified proxy).

- *Fixed $K$:* set $b = T/K$ (a re-parameterization) and optimize over $(\eta, \alpha)$.

### C.4 SHOULD YOU TUNE MOMENTUM?

Our results show that a rate of $T^{-1/4}$ can be achieved both with and without momentum tuning. This naturally raises the question: if we keep $\alpha$ fixed when moving from $T_0$ to $T_1 \gg T_0$, **how far are we from the token-optimal performance that one would obtain by re-tuning the momentum parameter** $\alpha$? We develop this in a few points, proving Corollary 1.

(1) PERFORMANCE GAP AT THE PROXY LEVEL IS ONLY A CONSTANT FACTOR (WHEN $b$ CAN SCALE). In the fixed-momentum large-horizon proxy (Eq. (4) in the main text),

$$\text{risk}_T(\eta, b; \alpha) \approx C_1 \frac{b}{\eta T} + \tilde{C}_2(\alpha) \frac{1}{\sqrt{b}} + \tilde{C}_3(\alpha)\eta, \qquad \tilde{C}_2(\alpha) = C_2\sqrt{\alpha}, \quad \tilde{C}_3(\alpha) = C_3\left(1 + \frac{1}{\alpha}\right).$$

Optimizing over $(\eta, b)$ for fixed $\alpha$ yields (App. C.1):

$$b_T^\star(\alpha) = \frac{\tilde{C}_2(\alpha)}{2\sqrt{C_1\tilde{C}_3(\alpha)}}\sqrt{T}, \tag{24}$$

$$\eta_T^\star(\alpha) = \frac{C_1^{1/4}\tilde{C}_2(\alpha)^{1/2}}{\sqrt{2}\,\tilde{C}_3(\alpha)^{3/4}} T^{-1/4}, \tag{25}$$

$$\text{risk}_T^\star(\alpha) = 2\sqrt{2}\,(C_1\tilde{C}_3(\alpha))^{1/4}\tilde{C}_2(\alpha)^{1/2}\,T^{-1/4}. \tag{26}$$

Substituting $\tilde{C}_2, \tilde{C}_3$ gives the explicit $\alpha$-dependence of the *constant*:

$$\text{risk}_T^\star(\alpha) = 2\sqrt{2}\,(C_1C_3)^{1/4}C_2^{1/2}\,(1 + \alpha)^{1/4}\,T^{-1/4}. \tag{27}$$

Thus, in this proxy regime, keeping $\alpha$ fixed does *not* change the exponent in $T$ (it remains $T^{-1/4}$), and re-tuning $\alpha$ **can only improve the *constant factor***. In fact, since $\alpha \in (0, 1]$, one has

$$(1 + \alpha)^{1/4} \in [1, 2^{1/4}],$$

so the maximal improvement from changing a fixed $\alpha$ is at most a factor $2^{1/4} \approx 1.19$ in performance. Equivalently, since $T$ enters as $T^{-1/4}$, this constant-factor improvement corresponds to at most a factor of 2 in token budget to reach a fixed target tolerance $\varepsilon$:

$$T_{\text{req}}(\alpha; \varepsilon) \propto (1 + \alpha) \cdot \varepsilon^{-4}.$$

For typical momentum values (e.g. $\beta = 0.9 \Rightarrow \alpha = 0.1$), this predicts only a mild potential gain in the proxy bound from re-tuning $\alpha$ at larger budgets.

(2) THE GAP CAN BE HUGE UNDER A BATCH-SIZE CAP: FIXED $\alpha$ CAUSES SATURATION. The previous conclusion assumes one can scale $b$ with $T$ as in equation 24. If instead the batch size is capped by hardware, $b \leq b_{\max}$, then the fixed-$\alpha$ proxy optimized over $\eta$ satisfies

$$\min_{\eta > 0} \text{risk}_T(\eta, b_{\max}; \alpha) \approx 2\sqrt{\frac{C_1\tilde{C}_3(\alpha)\, b_{\max}}{T}} + \tilde{C}_2(\alpha)\frac{1}{\sqrt{b_{\max}}}.$$

As $T \to \infty$, the first term vanishes but the second term remains:

$$\liminf_{T \to \infty} \min_{\eta > 0} \text{risk}_T(\eta, b_{\max}; \alpha) \gtrsim \frac{\tilde{C}_2(\alpha)}{\sqrt{b_{\max}}} = \frac{C_2\sqrt{\alpha}}{\sqrt{b_{\max}}}.$$

That is, *with fixed $\alpha$ and capped batch size, the proxy exhibits a non-vanishing noise floor.* By contrast, allowing $\alpha$ to decrease with $T$ (Regime (B) / Theorem 2 in the main text) removes this floor and restores $\text{risk}_T^\star \propto T^{-1/4}$ even at fixed $b$.

(3) WHAT IS LEFT ON THE TABLE IS PRIMARILY *batch growth* (REGIME (C) VS (D)). Comparing the token-optimal batch scaling under fixed $\alpha$ (Theorem 1: $b_T^\star \propto T^{1/2}$) to the jointly tuned scaling (Theorem 3 : $b_T^\star \propto T^{1/6}$) shows that re-tuning $\alpha$ mostly reduces the required batch growth with budget. A convenient way to quantify this is via the maximal budget that can be run *near-optimally* under a batch-size cap $b \leq b_{\max}$:

Regime (C): $b_T^\star \propto T^{1/2} \Rightarrow T \lesssim \Theta(b_{\max}^2)$, Regime (D): $b_T^\star \propto T^{1/6} \Rightarrow T \lesssim \Theta(b_{\max}^6)$,

(up to constant factors hidden in the proxy). Hence, even though both regimes achieve the same proxy exponent $T^{-1/4}$ in principle, *tuning momentum can dramatically extend the range of token budgets that remain feasible before hitting batch-size saturation.*

## C.5 GENERAL POWER-LAW SCALING UNDER A FIXED BUDGET

In this section, we prove Corollary 2. Recall that under the token (samples) budget $T = bK$ we can rewrite the bound as

$$\mathcal{U}_T(\alpha, \eta; b) := \mathcal{U}\left(\alpha, \eta; b, \frac{T}{b}\right) = \frac{b\Delta_0}{\eta T} + \frac{2\rho\sigma\sqrt{b}}{\alpha T} + 2\rho\sigma\sqrt{\frac{\alpha}{b}} + \frac{7L\eta}{2} + \frac{2L\eta}{\alpha}. \tag{28}$$

**Power-law schedules.** Assume that the algorithmic parameters follow power laws in $T$:

$$b(T) = \Theta(T^\beta), \qquad \alpha(T) = \Theta(T^{-\gamma}), \qquad \eta(T) = \Theta(T^{-\delta}), \tag{29}$$

with $\beta \in [0, 1]$ (since $1 \le b \le T$) and $\gamma, \delta \in \mathbb{R}$. *Note that here $\beta$ is not the momentum parameter!*

Then the five terms in equation 28 scale as

$$\frac{b\Delta_0}{\eta T} = \Theta(T^{\beta-1+\delta}) = \Theta\left(T^{-(1-\beta-\delta)}\right),$$

$$\frac{2\rho\sigma\sqrt{b}}{\alpha T} = \Theta\left(T^{\beta/2-1+\gamma}\right) = \Theta\left(T^{-(1-\beta/2-\gamma)}\right),$$

$$2\rho\sigma\sqrt{\frac{\alpha}{b}} = \Theta\left(T^{-(\beta+\gamma)/2}\right),$$

$$\frac{7L\eta}{2} = \Theta(T^{-\delta}),$$

$$\frac{2L\eta}{\alpha} = \Theta(T^{-\delta+\gamma}) = \Theta\left(T^{-(\delta-\gamma)}\right). \tag{30}$$

Equivalently, defining the decay exponents

$$r_1 := 1 - \beta - \delta, \quad r_2 := 1 - \frac{\beta}{2} - \gamma, \quad r_3 := \frac{\beta + \gamma}{2}, \quad r_4 := \delta, \quad r_5 := \delta - \gamma, \tag{31}$$

we have (up to absolute constants)

$$\mathcal{U}_T(\alpha(T), \eta(T); b(T)) = \Theta\left(T^{-r_1} + T^{-r_2} + T^{-r_3} + T^{-r_4} + T^{-r_5}\right) = \Theta\left(T^{-\min_i r_i}\right), \tag{32}$$

provided all $r_i > 0$ (i.e., each term decays).

**Guaranteeing a $T^{-1/4}$ bound for a prescribed batch scaling.** A convenient way to enforce a $T^{-1/4}$ rate is to equalize the three coupled terms $\frac{b}{\eta T}$, $\sqrt{\frac{\alpha}{b}}$, $\frac{\eta}{\alpha}$. Specifically, for any batch schedule $b(T)$ satisfying

$$b(T) \le c\sqrt{T} \qquad \text{(equivalently } \beta \le \tfrac{1}{2}\text{)}, \tag{33}$$

choose

$$\alpha(T) \propto \frac{b(T)}{\sqrt{T}}, \qquad \eta(T) \propto \frac{b(T)}{T^{3/4}}. \tag{34}$$

Then

$$\frac{b}{\eta T} = \Theta(T^{-1/4}), \qquad \sqrt{\frac{\alpha}{b}} = \Theta(T^{-1/4}), \qquad \frac{\eta}{\alpha} = \Theta(T^{-1/4}), \tag{35}$$

and the remaining terms satisfy

$$\frac{\sqrt{b}}{\alpha T} = \Theta\left(\frac{1}{\sqrt{bT}}\right) \le \Theta(T^{-1/2}), \qquad \eta = \Theta\left(\frac{b}{T^{3/4}}\right) \le \Theta(T^{-1/4}) \quad \text{(by equation 33)}. \tag{36}$$

Consequently,

$$\mathcal{U}_T(\alpha(T), \eta(T); b(T)) \lesssim C_1 T^{-1/4} + C_2 (bT)^{-1/2} = \mathcal{O}(T^{-1/4}), \tag{37}$$

for constants $C_1, C_2$ depending only on $\Delta_0, L, \rho, \sigma$ (and the hidden constants in equation 34).

**Remark (batch-size scaling $b = T/K$).** If $b(T) = \Theta(T^\beta)$, then $K(T) = T/b(T) = \Theta(T^{1-\beta})$ and $T/K = b = \Theta(T^\beta)$. Condition equation 33 is precisely $\beta \le \frac{1}{2}$ (i.e., $b$ cannot grow faster than $\sqrt{T}$) to maintain the $T^{-1/4}$ guarantee under power-law tuning.

**Aggressive batch-size growth: $\frac{1}{2} < \beta < 1$.** Assume the power-law batch schedule $b(T) = \Theta(T^\beta)$ with $\frac{1}{2} < \beta < 1$ (hence $K(T) = T/b(T) = \Theta(T^{1-\beta})$). In this regime, the bound cannot in general maintain the $T^{-1/4}$ decay: the two terms $\frac{b}{\eta T}$ and $\eta$ already impose a hard rate ceiling. Indeed, writing $\eta(T) = \Theta(T^{-\delta})$, their decay exponents are $r_1 = 1 - \beta - \delta$ and $r_4 = \delta$, so for any $\delta$,

$$\min\{r_1, r_4\} \;\leq\; \max_\delta \min\{1 - \beta - \delta, \delta\} \;=\; \frac{1 - \beta}{2}, \tag{38}$$

with equality achieved by balancing them at

$$\delta^\star = \frac{1 - \beta}{2} \qquad \Longleftrightarrow \qquad \eta(T) = \Theta\left(T^{-(1-\beta)/2}\right). \tag{39}$$

Taking, e.g., $\alpha(T) = \Theta(1)$ and $\eta(T)$ as in equation 39 yields

$$\mathcal{U}_T\big(\alpha(T), \eta(T); b(T)\big) = \Theta\left(T^{-(1-\beta)/2}\right), \qquad \frac{1}{2} < \beta < 1, \tag{40}$$

since the remaining terms decay strictly faster:

$$\frac{\sqrt{b}}{\alpha T} = \Theta\left(T^{-(1-\beta/2)}\right), \qquad \sqrt{\frac{\alpha}{b}} = \Theta\left(T^{-\beta/2}\right), \qquad \frac{\eta}{\alpha} = \Theta\left(T^{-(1-\beta)/2}\right).$$

Equivalently, using $K(T) = T/b(T) = \Theta(T^{1-\beta})$, the achievable rate can be written as

$$\Theta\left(T^{-(1-\beta)/2}\right) \;=\; \Theta\left(K^{-1/2}\right) \;=\; \Theta\left(\sqrt{\frac{b}{T}}\right), \tag{41}$$

highlighting that when $b$ grows faster than $\sqrt{T}$, the bound becomes iteration-limited.

## C.6 ANALYSIS OF SGD AND COMPARISON WITH LMO METHODS

For context, consider *standard* Euclidean SGD (not normalized SGD / LMO),

$$x^{k+1} = x^k - \eta\, g_b^k, \qquad \mathbb{E}[g_b^k \mid x^k] = \nabla f(x^k), \qquad \mathbb{E}\|g_b^k - \nabla f(x^k)\|_2^2 \leq \sigma^2/b,$$

with $f$ being $L$-smooth in $\|\cdot\|_2$. A classical non-convex guarantee (for constant $\eta \leq 1/L$) is [17]

$$\frac{1}{K}\sum_{k=0}^{K-1} \mathbb{E}\|\nabla f(x^k)\|_2^2 \;\lesssim\; \frac{\Delta_0}{\eta K} + \frac{L\eta\sigma^2}{b}. \tag{42}$$

Equivalently, $\min_{0 \leq k \leq K-1} \mathbb{E}\|\nabla f(x^k)\|_2^2$ obeys the same RHS up to constants.

**Fixed K.** Optimizing equation 42 over $\eta$ yields $\eta^\star \propto \sqrt{b/K}$ and

$$\min_\eta \frac{1}{K}\sum_{k=0}^{K-1} \mathbb{E}\|\nabla f(x^k)\|_2^2 \;\lesssim\; \sqrt{\frac{\Delta_0 L\sigma^2}{bK}}.$$

Thus, at a fixed iteration budget $K$, increasing $b$ improves the bound (as in many stochastic methods).

**Fixed token budget $T = bK$.** Substituting $K = T/b$ into equation 42 gives

$$\frac{1}{K}\sum_{k=0}^{K-1} \mathbb{E}\|\nabla f(x^k)\|_2^2 \;\lesssim\; \frac{\Delta_0\, b}{\eta T} + \frac{L\eta\sigma^2}{b}.$$

Optimizing over $\eta$ ($\eta^\star \propto bT^{-1/2}$) yields an optimized value

$$\min_\eta \left\{\frac{\Delta_0\, b}{\eta T} + \frac{L\eta\sigma^2}{b}\right\} \;\lesssim\; \sqrt{\frac{\Delta_0 L\sigma^2}{T}},$$

which is *independent of $b$* at the level of this proxy bound. In other words, under a fixed token budget, the *classical SGD analysis does not predict a non-trivial token-optimal batch size*: after optimizing $\eta$, $b$ largely trades off iterations vs. variance reduction in a way that cancels.

**Comparison to LMO.** The LMO bound used in the paper (equation 10) differs structurally: (i) it controls $\|\nabla f(\cdot)\|_\star$ (not squared), and (ii) it contains additional terms coupling $(\eta, \alpha)$ and mini-batching, including a noise-floor term $\propto \sigma\sqrt{\alpha/b}$ and a burn-in / initialization term that can scale as $\propto \sigma/(\alpha\sqrt{b}K)$ under matched stochastic initialization (Appendix D.1). As a consequence,

- with *fixed* $\alpha$, the proxy retains a decreasing-in-$b$ contribution under $T = bK$ (e.g. $\propto 1/\sqrt{b}$), leading to a genuinely non-trivial token-optimal batch size (Theorem 1: $b_T^\star \propto \sqrt{T}$);

- with *tuned* $\alpha$, the leading-order cancellation in $b$ under $T = bK$ becomes more similar in spirit to SGD, and $b^\star(T)$ is pinned only by lower-order terms (Theorem 2).

# D  HOW MODIFIED ASSUMPTIONS CAN CHANGE THE EXPONENTS

## D.1  DEPENDENCY ON INITIALIZATION

**Matched initialization and the burn-in term.** We assume *matched* initialization, namely that the momentum buffer is initialized from a stochastic mini-batch gradient of the *same* batch size $b$ as used during training, e.g. $m^0 = g_b^0$. Under the bounded-variance model, this implies

$$E_0 := \mathbb{E}\big[\|g_b^0 - \nabla f(x^0)\|_\star\big] \lesssim \frac{\rho\sigma}{\sqrt{b}},$$

and this is exactly what yields the $\frac{1}{\alpha\sqrt{b}K}$ "burn-in" dependence in the non-convex LMO bound below. If $m^0$ is not matched, this term changes; see next paragraph.

**On non-matched initialization.** If $m^0$ is *not* initialized from a stochastic batch-$b$ gradient, the burn-in term can change from $\propto \frac{1}{\alpha\sqrt{b}K}$ to $\propto \frac{E_0}{\alpha K}$ with an $E_0$ that may not decay as $1/\sqrt{b}$. Under a fixed budget $T = bK$ this changes the $\sqrt{b}/(\alpha T)$ structure in equation 11 and can alter the token-optimal $b(T)$ scaling.

## D.2  NOISE MODEL AND FLATNESS OF THE BATCH SIZE LANDSCAPE

The power-law exponents obtained by optimizing convergence upper bounds are not universal: they are a direct consequence of (i) the functional form of the bound and (ii) how the stochastic error term scales with mini-batching, geometry, and moment assumptions. We record a simple sensitivity analysis that makes the dependence explicit.

I. A ONE-PARAMETER MODEL FOR HOW NOISE SHRINKS WITH BATCH SIZE. The equation 2 bound assumes a *finite-variance* (sub-Gaussian/sub-exponential) scaling, where the typical noise magnitude decreases as $b^{-1/2}$. To capture deviations (e.g. correlations or heavy tails), we introduce a generic exponent $q$:

**Assumption 1 (Effective mini-batch noise scaling)** *There exist $q \in (0, 1]$ and a scale parameter $\sigma_q > 0$ such that the mini-batch gradient satisfies*

$$\mathbb{E}\big[\|g_b(x) - \nabla f(x)\|_\star\big] \lesssim \frac{\sigma_q}{b^q} \qquad \text{(uniformly over } x\text{)}.$$

*The bounded-variance/i.i.d. setting corresponds to $q = \frac{1}{2}$. Correlations or other inefficiencies can lead to $q < \frac{1}{2}$.*

Under Assumption 1, the two "noise" terms in the LMO bound scale as

$$\frac{1}{\alpha K} \cdot \frac{\sigma_q}{b^q} \qquad \text{and} \qquad \sqrt{\alpha} \cdot \frac{\sigma_q}{b^q},$$

rather than with $b^{-1/2}$.

II. WHY THE LEADING-ORDER BOUND CAN BECOME "FLAT" IN $b$ (AND WHEN IT DOES NOT). To isolate the mechanism, consider the dominant three-term proxy (dropping constants and the burn-in term for the moment)

$$\mathcal{U}(\alpha, \eta; b, K) \approx \frac{\Delta_0}{\eta K} + L\frac{\eta}{\alpha} + \frac{\sigma_q}{b^q}\sqrt{\alpha}. \qquad (43)$$

Optimizing equation 43 over $\eta$ gives

$$\eta^\star(\alpha) \propto \sqrt{\frac{\Delta_0\,\alpha}{L\,K}}, \qquad \min_{\eta>0}\left\{\frac{\Delta_0}{\eta K} + L\frac{\eta}{\alpha}\right\} \propto \sqrt{\frac{\Delta_0 L}{K}} \cdot \alpha^{-1/2}.$$

Then the $\alpha$-problem becomes $c_1\alpha^{-1/2} + c_2(b)\alpha^{1/2}$ with $c_2(b) \propto \sigma_q/b^q$, hence

$$\alpha^\star(b, K) \propto \frac{b^q}{\sqrt{K}}, \qquad \eta^\star(b, K) \propto \frac{b^{q/2}}{K^{3/4}}, \qquad \min_{\alpha,\eta}\mathcal{U} \propto \frac{b^{-q/2}}{K^{1/4}}. \qquad (44)$$

Now impose a fixed token budget $T = bK$ (so $K = T/b$). Plugging into equation 44 yields

$$\min_{\alpha,\eta}\mathcal{U} \propto T^{-1/4}b^{1/4-q/2}. \qquad (45)$$

**Interpretation of equation 45.**

- If $q = \frac{1}{2}$ (bounded variance, i.i.d. mini-batching), then $1/4 - q/2 = 0$ and *the leading-order dependence on $b$ cancels.* This is precisely the "flatness in $b$" phenomenon: once $\eta$ and $\alpha$ are retuned for each $b$, the dominant term depends primarily on $T = bK$ rather than on $b$ itself.

- If $q < \frac{1}{2}$ (noise shrinks *slower* than $b^{-1/2}$), then $1/4 - q/2 > 0$ and the leading term *increases* with $b$; the token-optimal batch size is pushed toward the smallest feasible $b$.

- If $q > \frac{1}{2}$ (noise shrinks *faster* than $b^{-1/2}$), then $1/4 - q/2 < 0$ and larger batches become beneficial already at leading order.

Therefore, the existence (and scaling) of an *interior* token-optimal batch size is *not robust*: it relies on the finite-variance $q = \frac{1}{2}$ law, plus lower-order terms (e.g. burn-in) that break the leading-order cancellation. This explains why the joint optimum $b_T^\star = \Theta(T^{1/6})$ should not be read as the only viable scaling: for bounded-variance noise, many choices of $b(T)$ remain near-optimal once $\alpha(T)$ is tuned (momentum compensates for smaller batches), and the burn-in term selects $b_T^\star$ only through lower-order effects.

III. HEAVY-TAILED NOISE: BOTH THE $b$-LAW AND THE $T$-EXPONENT CAN CHANGE. A common empirical observation in deep learning is that stochastic gradient noise can be heavy-tailed, often modeled via $\alpha$-stable laws [38, 45]; in such regimes the variance may be infinite and the classical $b^{-1/2}$ scaling can fail. In a stylized $\mathfrak{p}$-moment model with $\mathfrak{p} \in (1, 2)$, a typical scaling for sample averages is

$$\text{noise magnitude} \propto b^{-(1-1/\mathfrak{p})}, \quad \text{i.e.} \quad q = 1 - \frac{1}{\mathfrak{p}} < \frac{1}{2}.$$

Plugging into equation 45 gives

$$\min_{\alpha,\eta}\mathcal{U} \propto T^{-1/4}b^{-1/4+1/(2\mathfrak{p})},$$

which increases with $b$ for any $\mathfrak{p} < 2$, again pushing the token-optimal $b$ toward small batches.

Moreover, under heavy-tailed noise the *optimal* convergence rate in $T$ can itself differ from $T^{-1/4}$ (the finite-variance case), and depends on $\mathfrak{p}$. This indicates that mismatches between empirical scaling-law fits and finite-variance theory can reflect a genuinely different regime rather than just loose constants.

IV. "VARIANCE" IN NON-EUCLIDEAN METHODS: WHICH NORM MATTERS. For LMO/Muon, the stochastic terms are naturally expressed in the dual norm $\|\cdot\|_\star$. Accordingly, a more intrinsic noise proxy is

$$\sigma_\star^2 := \sup_x \mathbb{E}\big[\|g(x) - \nabla f(x)\|_\star^2\big],$$

rather than an $\ell_2$-variance. Using norm compatibility one can always upper bound $\|v\|_\star \leq \rho\|v\|_2$ and therefore $\sigma_\star \leq \rho\sigma_2$, but this can be loose and may hide dimension/model-size dependence in $\rho$ (or in $\sigma_\star$ itself). Hence, changing the *noise model* from an $\ell_2$ variance bound to a $\|\cdot\|_\star$-variance bound can change effective constants and, when combined with model-size scaling, can affect fitted exponents.

V. PARAMETER CONSTRAINTS ALSO CHANGE APPARENT EXPONENTS. All of the above assumes $\eta$ and $\alpha$ can be freely retuned as $T$ varies. In practice, hyperparameters are constrained (e.g. $\alpha$ fixed, $\eta$ capped for stability, discrete grids). Such constraints remove the cancellation behind equation 45 and can lead to different effective power laws (e.g. a "critical batch size" beyond which increasing $b$ is harmful because $\eta$ cannot be increased accordingly).

### D.3 WHY CAN EMPIRICAL FITS SHOW $\eta^\star$ INCREASING WITH TOKEN BUDGET?

Several empirical works report a one-dimensional fit $\eta^\star(T) \propto T^q$ while scaling the token budget via $T = bK$. In contrast, our proxy bound is multi-variate and depends on $(b, K, \alpha, \eta)$, so an "$\eta$ vs. $T$" exponent is *not intrinsic*: it is *conditional on the scaling path* $T \mapsto (b(T), K(T), \alpha(T), \eta(T))$.

**A protocol identity for effective exponents.** Assume that over the (finite) range probed in practice, the tuned constant step size can be approximated by a separable power law

$$\eta^\star(b, K) \propto b^\kappa K^{-\lambda}, \qquad \kappa \geq 0, \ \lambda \geq 0. \tag{46}$$

Along any batch-growth path $b(T) \propto T^p$ (hence $K(T) = T/b(T) \propto T^{1-p}$), equation 46 implies an *effective* token exponent

$$\eta^\star(T) \propto T^{q_{\text{eff}}}, \qquad q_{\text{eff}} = \kappa p - \lambda(1-p). \tag{47}$$

In particular,

$$q_{\text{eff}} > 0 \iff p > \frac{\lambda}{\kappa + \lambda}. \tag{48}$$

Thus, even if $\eta^\star$ decreases with the horizon at fixed batch ($\lambda > 0$), a positive fitted exponent in $T$ can arise if the batch-growth effect dominates.

**Instantiations for our LMO proxy.** *(i) Fixed momentum proxy.* Optimizing the proxy over $\eta$ at fixed $(b, K, \alpha)$ gives $\eta^\star(b, K) \propto K^{-1/2}$, i.e. $(\kappa, \lambda) = (0, \frac{1}{2})$. Hence $q_{\text{eff}} = -(1-p)/2 \leq 0$ for any $p$: in the fixed-momentum proxy, $\eta^\star(T)$ cannot increase with $T$.

*(ii) Fixed-$b$ $K$-optimal schedules evaluated along $T = bK$.* Minimizing the LMO proxy at fixed $b$ yields (up to constants / lower-order terms)

$$\eta^\star(b, K) \propto b^{1/4} K^{-3/4}, \qquad \alpha^\star(b, K) \propto \sqrt{b/K}.$$

Thus $(\kappa, \lambda) = (\frac{1}{4}, \frac{3}{4})$ and equation 47 gives

$$q_{\text{eff}} = p - \tfrac{3}{4}. \tag{49}$$

Therefore, a positive fitted exponent is possible under batch-heavy paths $p > 3/4$.

**Caveat: saturation of $\alpha \leq 1$.** The schedule above also implies $\alpha^\star(T) \propto b/\sqrt{T}$. Along $b(T) \propto T^p$, we have $\alpha^\star(T) \propto T^{p-1/2}$, so for $p > 1/2$ one eventually reaches a regime where $\alpha$ saturates at 1, and the effective $\eta$ scaling transitions away from equation 49.

**Connection to empirical scaling laws.** This "path-conditioning" mechanism can explain *why* a study may report $\eta^\star$ increasing with $T$ even though the globally token-optimal schedules in our analysis yield a decreasing $\eta^\star(T)$.

However, it does not guarantee matching exponents. For example, StepLaw reports (at fixed model size) $b^\star(T) \propto T^{0.571}$ and $\eta^\star(T) \propto T^{0.307}$ [26], i.e. $p \simeq 0.571 < 3/4$ but $q \simeq 0.307 > 0$ [26]. This cannot be explained by the LMO $K$-optimal proxy exponent $q_{\mathrm{eff}} = p - \frac{3}{4}$ unless the effective exponents $(\kappa, \lambda)$ in equation 46 differ substantially from $(\frac{1}{4}, \frac{3}{4})$, or the proxy assumptions fail.

As another reference point, von Rütte et al. [42] report $b^\star(T) \propto T^{0.8225}$ and $\eta^\star \propto (b^\star)^{0.3412}$ for diffusion LMs, implying $\eta^\star(T) \propto T^{0.28}$ over their explored range [42]. The batch exponent $p > 3/4$ falls in the batch-heavy regime where positive $q_{\mathrm{eff}}$ is possible, but the magnitude still differs from the LMO proxy, suggesting additional effects. So $p - 3/4$ is best viewed as one concrete mechanism (under a specific proxy + tuning protocol), not a general prediction for empirical scaling.

## E  PLOTS

**Plots referenced in the main paper.** Due to space constraints, we refer to these here.

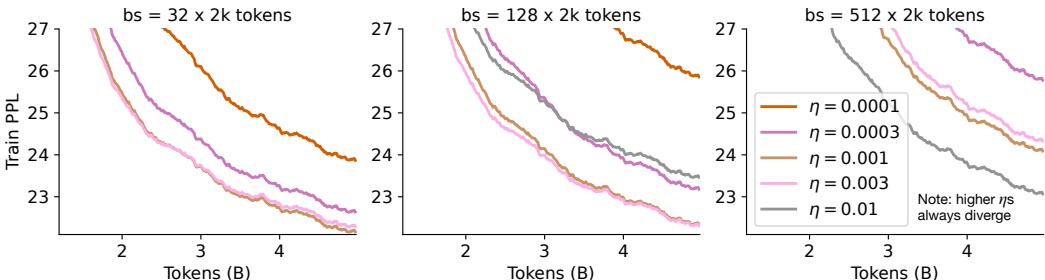

Figure 1: We perform a constant-$\eta$ training experiment on a 160M transformer (PlainLM implementation [1]), trained with a language modeling objective for up to $5B$ tokens from SlimPajama [40]. The setup is the same as in Theorem 1 and Figure 2. With $8\%$ warmup + constant learning rate (Adam betas fixed to $(0.9, 0.95)$), grid-searching $\eta$ across batch sizes shows clear structure: (i) at fixed batch size, the optimal $\eta$ decreases over training time (e.g., $b = 32$ shifts from 0.003 to 0.001 after $2.5B$ tokens); (ii) this trend appears similarly for $b = 128$, albeit the switching point is not yet reached at $5B$ tokens; and (iii) the optimal $\eta$ increases with batch size (at $T = 5B$ tokens, $\eta = 0.001$ at $b = 32$, $\eta = 0.003$ at $b = 128$, $\eta = 0.01$ at $b = 512$). Constant $\eta$ allows clean finite-time comparisons that reveal both token-dependent and batch-dependent optimal $\eta$s.

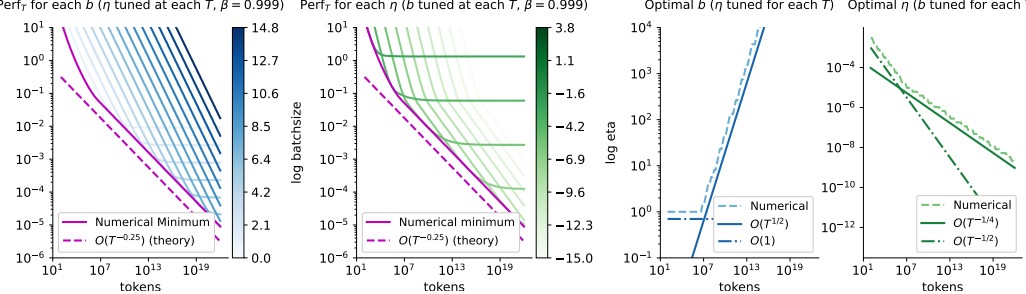

Figure 2: Verification of Theorem 1. Shown are the trends of equation 4 ($C_1 = C_2 = C_3 = 1$) under the choice $\beta = 1 - \alpha = 0.999$. More $\alpha$s can be found in App. E. In the two plots on the left, we show in magenta performance at the best value of $(\eta, b)$ for each token budget, following $\mathcal{O}(T^{-1/4})$. Plotted in blue are also performances for a fixed batch size, minimizing over $\eta$ at each token budget, and in green performances for a fixed learning rate, minimizing $b$ at each token budget. In the two plots on the right, we show how the optimal batch size and learning rate scale with tokens. The trends predicted by Theorem 1 hold after a burn-in phase, where the optimal batch size is $b = 1$.

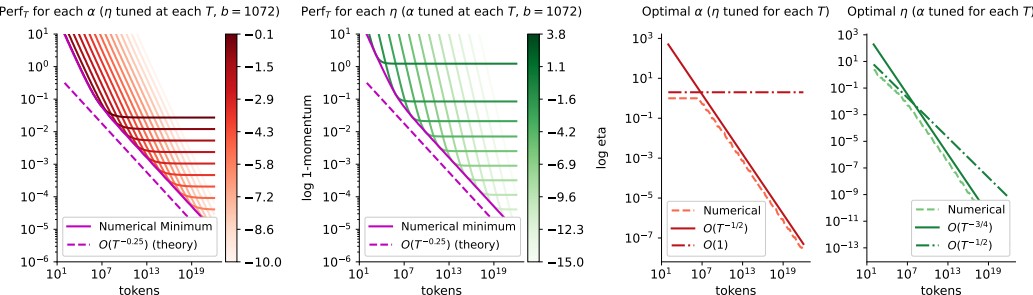

Figure 3: Numerical verification of Theorem 2 for $b = 1072$. The setting is same as for Figure 2.

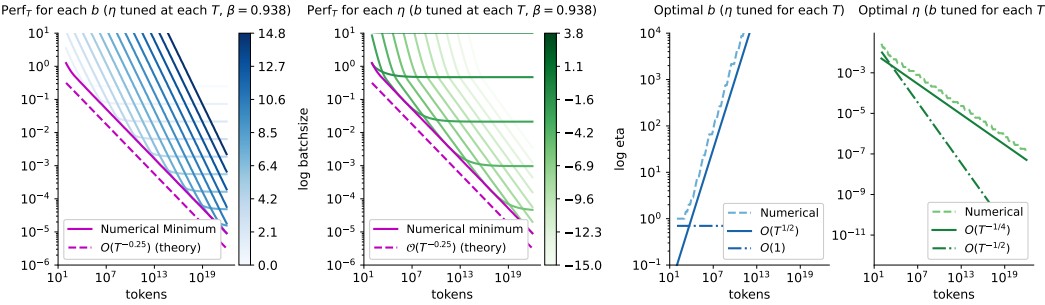

Figure 4: Numerical verification of Theorem 1 for $\beta = 0.934$.

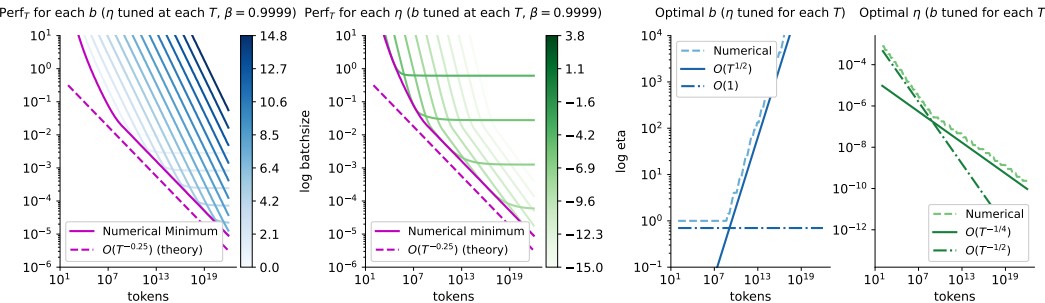

Figure 5: Numerical verification of Theorem 1 for $\beta = 0.9999$.

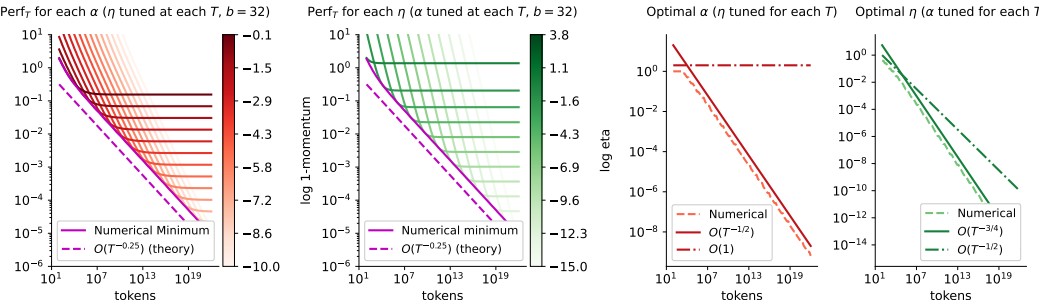

Figure 6: Numerical verification of Theorem 2 for $b = 32$.

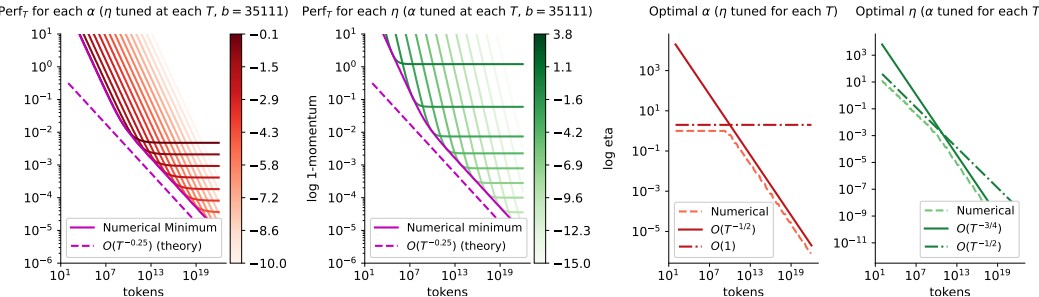

Figure 7: Numerical verification of Theorem 2 for $b = 35111$.

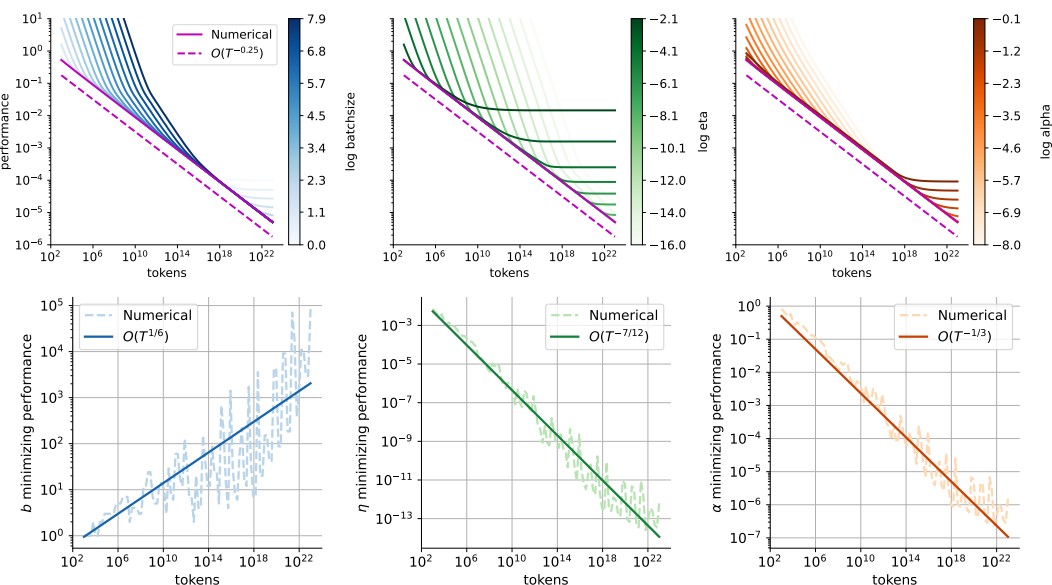

Figure 8: Numerical verification of Theorem 3. Oscillations are due to the grid search and to similar performance among hyperparameter choices. See Appendix D.2 for further comments.

**Iso-loss curves.**   As shown in Figure 9, under fixed momentum, achieving a certain target performance requires a minimum batch size. We identify a trend of $b_T^\star \propto T^{1/2}$ as predicted by our theory. The saturation of the near-optimal line in the left panel indicates that, at large token budgets, performance becomes less sensitive to batch size scaling when the learning rate $\eta$ is lower bounded, since smaller momentum requires proportionally smaller $\eta$.

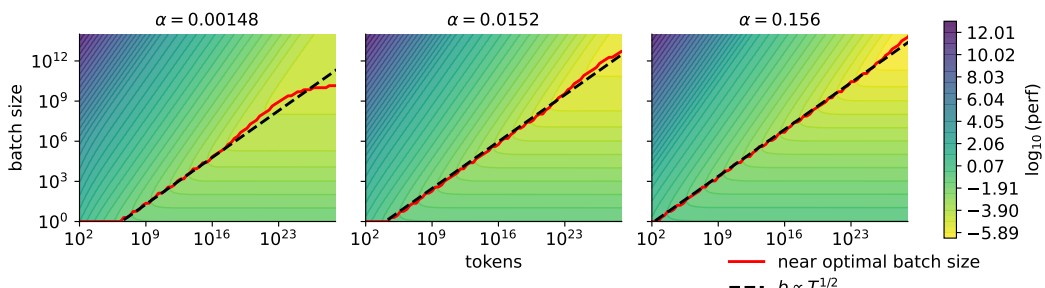

Figure 9: Contours of best achievable performance versus batch size and number of tokens. Fixed $\alpha$ at different value, tuned $\eta \in [1.7e{-}6, 1]$. The red curve denotes the near-optimal (99% performance) batch size as a function of the number of tokens.

**Three regimes and a hyperbolic $(K, \sqrt{b})$ tradeoff (iso-performance curves).**   Fix a target level $\varepsilon$ and let $c := \varepsilon - \text{const} > 0$ absorb terms independent of $(b, K)$. Assuming the learning rate is well tuned for each $(b, K)$ (i.e., $\eta$ minimizes the $\Delta_0/(\eta K) + L\eta(\frac{7}{2} + \frac{2}{\alpha})$ part of the bound), the remaining dependence on $(b, K)$ can be summarized as

$$u_\eta(b, K) \approx \frac{C_{\text{det}}}{\sqrt{K}} + \frac{C_{\text{burn}}}{K\sqrt{b}} + \frac{C_{\text{floor}}}{\sqrt{b}},$$

where $C_{\text{det}} := 2\sqrt{\Delta_0 L(\frac{7}{2} + \frac{2}{\alpha})}$, $C_{\text{burn}} := \frac{2\rho\sigma}{\alpha}$, $C_{\text{floor}} := 2\rho\sigma\sqrt{\alpha}$.

Setting $u_\eta(b, K) = c$ reveals three regimes:

1. Iteration-limited ($b \to \infty$) with $K_{\min} = (C_{\text{det}}/c)^2$;

2. Batch-limited ($K \to \infty$) with $b_{\min} = (C_{\text{floor}}/c)^2$;

3. An intermediate tradeoff region where the burn-in term dominates, yielding $K\sqrt{b} \approx C_{\text{burn}}/c$, i.e. $\sqrt{b} \propto 1/K$.

We can also rearrange the equation into:

$$\left(c\sqrt{K} - C_{\text{det}}\right)\left(c\sqrt{b} - \left(C_{\text{floor}} + \frac{C_{\text{burn}}}{K}\right)\right) = C_{\text{det}}\left(C_{\text{floor}} + \frac{C_{\text{burn}}}{K}\right) \approx C_{\text{det}}\left(C_{\text{floor}} + \frac{C_{\text{burn}}}{K_0}\right)$$

This mirrors the shifted-hyperbola iso-loss law reported by von Rütte et al. [42] (see their Eq. (7)), up to our use of $\sqrt{b}$ induced by the $1/\sqrt{b}$ noise scaling.

When momentum is fixed, and learning rate is tuned, the hyperbolic relationship between batch size and iterations is shown in the middle panel of Figure 10. Note that on the left panel, if $\alpha$ and $\eta$ can be unrestrictedly tuned, the final performance achieved is higher than that of the middle and right panels. This is because a larger batch size requires a much smaller learning rate, which is often not realistic. If the $\eta$ grid is lower bounded, then there is no significant difference in the achievable performance between fixed $\alpha$ (middle) and jointly tuned $\alpha$ (right).

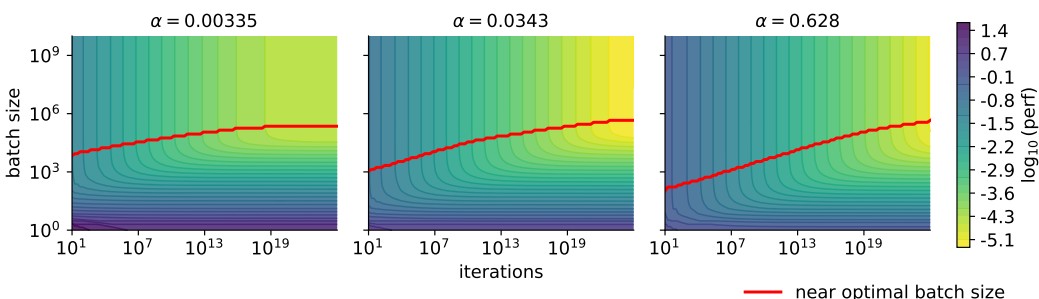

Figure 10: Contours of best achievable performance versus batch size and training iterations. Fixed $\alpha$ at various values, and tuned $\eta \in [1.07e - 7, 1]$. The red curve denotes the near-optimal (99% performance) batch size as a function of the number of iterations.

