# OpenReview forum: "Deriving Hyperparameter Scaling Laws via Modern Optimization Theory"
_ICLR.cc/2026/Workshop/Sci4DL — Sci4DL 2026_

### Official Review · Reviewer_C3qw · 2026-02-25

**Fit:** 2
**Significance:** 2
**Confidence:** 3

**Summary:**

This paper derives closed-form power-law schedules for learning rate, momentum, and batch size as functions of either iteration steps or token budget, by treating a modern non-convex convergence upper bound for LMO-based optimizers (normalized SGD, signSGD, Muon)  and minimizing it under different tuning regimes.

**Strengths:**

Key takeaway is a token-optimal batch size $b\sim T^{1/2}$ under fixed momentum and token budget $T$. Similar scaling has been observed in empirical studies of large language models. The result is relevent to modern optimizers beyond plain SGD.

**Suggestions:**

Limited empirical validation should be improved. The “verification” largely checks that numerical minimization matches the derived exponents, rather than demonstrating that these schedules are optimal in actual deep-learning training runs.

---

### Official Review · Reviewer_zpkb · 2026-02-26

**Fit:** 2
**Significance:** 2
**Confidence:** 2

**Summary:**

The paper theoretically derives joint scaling laws for LR, batch size, and momentum as a function of token budget by utilizing convergence analysis for LMO-based optimizers. They minimize the theoretical convergence bound to derive the joint scaling laws and validate them numerically.

**Strengths:**

Solid theoretical analysis of joint scaling laws of hparams.

**Suggestions:**

* The discrepancy between the theory and empirical findings of learning rate scaling can benefit from further discussion.
* The paper can benefit from experiments
* How do the results change with the LR schedule and model size?

---

### Official Review · Reviewer_WybU · 2026-02-27

**Fit:** 3
**Significance:** 2
**Confidence:** 2

**Summary:**

This paper derives scaling laws for key training hyperparameters (learning rate, momentum, and batch size) by treating a convergence bound (for LMO-based optimizer variants) as a proxy objective and optimizing it with respect to training budget. It yields closed-form power-law schedules and “budget transfer” rules that predict how near-optimal hyperparameters should change as token/iteration budget increases, including the emergence of a token-optimal (critical) batch size under certain regimes. The authors then numerically validate their derivations by directly minimizing the proxy bound and visualizing the resulting optima across budgets.

**Strengths:**

* The paper is generally clear and well-written, with an easy-to-follow presentation (assumptions to bound to derived scalings and numerical verification)
* The authors are transparent about limitations and simplifying assumptions (e.g., proxy/bound-based empirics, constants set to 1, omissions like weight decay and fixed learning rates)
* The authors directly verify the theoretical derivations by numerically optimizing the same proxy bound and useful visual intuition via contour plots.

**Suggestions:**

* I wonder if the authors have done a small-scale real training study to test whether the predicted batch/LR scalings and transfer rules hold beyond proxy minimization. As the authors note "our results suggest that [emergence of a critical batch size] may arise from a restricted learning rate range whereas the empirical literature often connects CBS to gradient noise", could this be verified by empirically separating learning-rate ceiling effects from gradient-noise-scale effects (e.g., 2D sweeps over batch size x learning rate)?
* (more minor) Figures are quite dense and would benefit from more informative captions

---

### Meta-Review · Area_Chair_oAnN · 2026-02-28

**Recommendation:** Accept

**Metareview:**

This paper studies the scaling of optimization hyperparameters with number of gradient steps and data size. All reviewers noted the importance of the topic and soundness of the results. I recommend acceptance.

---

### Decision · Program_Chairs · 2026-03-02

Accept